# Monkey See, Model Knew

## LARGE LANGUAGE MODELS ACCURATELY PREDICT HUMAN + MACAQUE VISUAL BRAIN ACTIVITY

Colin Conwell[*1], Emalie MacMahon[1], Kasper Vinken[2], Saloni Sharma[2], Akshay Jagadeesh[2], Jacob S. Prince[3], George A. Alvarez[3], Talia Konkle[3], Leyla Isik[1], and Margaret Livingstone[2]

[1]Johns Hopkins University, Department of Cognitive Science
[2]Harvard Medical School, Department of Neurobiology
[3]Harvard University, Department of Psychology

### ABSTRACT

Recent progress in multimodal AI and "language-aligned" visual representation learning has rekindled debates about the role of language in shaping the human visual system. In particular, the emergent ability of "language-aligned" vision models (e.g. CLIP) – and even pure language models (e.g. BERT) – to predict image-evoked brain activity has led some to suggest that human visual cortex itself may be "language-aligned" in comparable ways. But what would we make of this claim if the same procedures worked in the modeling of visual activity in a species that does not have language? Here, we deploy controlled comparisons of pure-vision, pure-language, and multimodal vision-language models in prediction of human (N=4) and rhesus macaque (N=6, 5:IT, 1:V1) ventral visual activity evoked in response to the same set of 1000 captioned natural images (the "NSD1000"). The results reveal markedly similar patterns in aggregate model predictivity of early and late ventral visual cortex across both species. This suggests that language model predictivity of the human visual system is not necessarily due to the evolution or learning of language *per se*, but rather to the statistical structure of the visual world that is reflected in the statistics of language *as data*.

## 1 INTRODUCTION

The idea that language shapes how we 'see' the world has long been one of the most actively debated ideas in cognitive (neuro)science (1; 2; 3; 4; 5), and has evolved through many forms over that time. A recent evolution of this idea has manifested in the form of competing hypotheses about the extent to which high-level human visual cortex is 'language-aligned' – or, in other words, the extent to which lingustic or linguistically-learned structure is evident in visual brain responses (6; 7). The resurgence of this debate is predicated in large part on two seminal findings in research on modern deep learning models: first, the finding that 'language-aligned' machine vision models (e.g. CLIP) are some of the most predictive models to date of image-evoked activity in the visual brain (8); and second, the finding that even pure-language models (e.g. BERT) are capable of predicting image-evoked brain activity by way of image captions alone (9; 10).

Here, we apply a logical razor to this debate in the form of assessing whether these two key findings hold in the brain of a species that does not have language. We call this the 'monkey razor', and define it as follows: If the ability of 'language-aligned' vision models or pure-language models to predict image-evoked brain activity is indeed evidence of language having (re-)shaped visual representation, we should not find similar predictivity in monkeys.

Our approach is to use encoding models fit to the feature spaces of a diverse set of pure-vision, pure-language (LLMs), and multimodal (language-aligned) vision (VLMs) models to predict image-evoked brain activity in the ventral stream of 4 humans and 6 rhesus macaques shown the same set of 1000 natural images from the Natural Scenes Dataset (NSD) (11). The brain-likeness of the pure-vision and language-aligned vision models is assessed on the images themselves. The brain-likeness of the pure-language models is assessed using an average of the embeddings for the first 5 captions

---

[*]Correspondence: conwell@g.harvard.edu

associated with each image (collected from metadata for the MS-COCO dataset (12), from which NSD images are curated).

We find that vision and language models provide accurate predictions of neural responses in high-level ventral stream regions of both human and macaque visual cortex. The majority of variance explained by pure-vision and pure-language models in human OTC is shared with macaque IT, and the remaining 'uniquely human' variance is more or less equally distributed across the two model modalities. Together, these results suggest that language model predictivity of human visual cortex is likely not a function of language-like representation injected into vision by human-like language learning, and instead a reflection of a convergence between vision and language models based most directly on the large-scale, end-to-end statistical learning that defines them both.

## 2 RESULTS

**General Approach** Our encoding procedure follows an established protocol for large-scale model comparison (13), and includes a nested cross-validation regime that decontaminates the selection of the most brain-like layer *within* each of our candidate models (assessed on a 'training set' of 500 images) from the comparison of brain-likeness *between* models (assessed on the held-out 'test set' of 500 images). The predictivity of the encoding models is assessed with the raw Pearson correlation ($r$) between model-predicted and actual brain activity. (The calculation of comparable noise ceilings across species for use in a measure of 'explainable variance explained' (14; 15) is an area of ongoing investigation. Here, for illustration, we use the same noise-corrected signal-to-noise-ratio (NCSNR) calculation used in the NSD (11)).

The encoding models for the human (fMRI) brain activity are fit to reliability-selected voxels ($NCSNR > 0.2$) in a broad mask of early visual cortex (EVC, N=15326 voxels) and occcipitem-poral cortex (OTC, N=29840 voxels), with both anatomical and functional criteria as the basis of inclusion. The encoding models for the monkey (electrophysiology) brain activity are fit to multi-unit responses (i.e. average firing rates in a 150ms window) from arrays placed either in macaque V1 (N=34 units) or inferotemporal (IT) cortex (N=394 units). Note that we use the following convention for reporting statistics: statistic [lower, upper] 95% (bootstrapped) confidence interval.

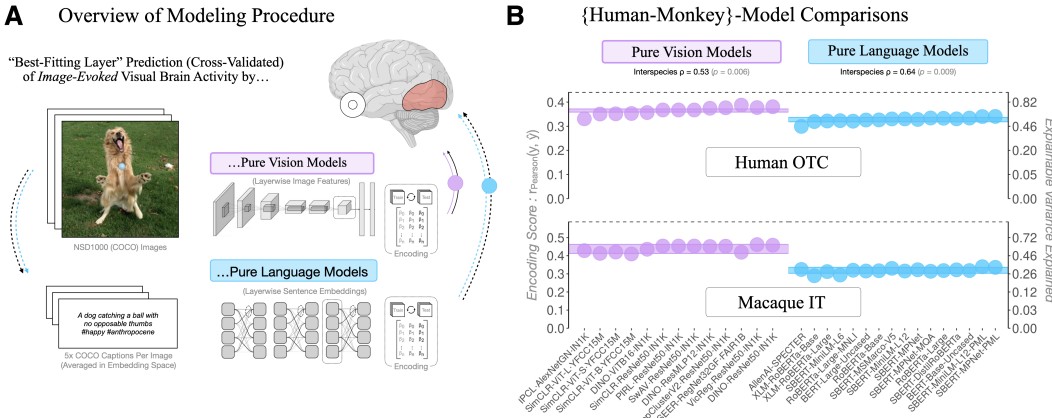

Figure 1: (A) Overview of modeling procedure: *Image-evoked visual brain activity* is predicted by way of image features (for vision models) or the embeddings of image captions (for language models), using a unit-wise (voxel / neuron) encoding analysis. (B) Encoding accuracies from the most brain-like layer of a series of (unimodal) pure vision and pure language models in prediction of both human occipitotemporal cortex (OTC) and macaque inferotemporal cortex (IT). Indiviudal points are the accuracies for individual subjects (human or macaque). The vertical striped boxes are the means ±95%CIs (across subjects) per model. The horizontal, semitranslucent rectangles extending over these striped boxes are the means ±95% CIs (across models) per model type (modality).

**Vision versus Language in Human Ventral Stream** Commensurate with previous findings (16; 9), we find that pure-language model embeddings over image captions are sufficient to predict high-

level human ventral visual activity almost as accurately as pure-vision models (Figure 1B: Top Row, Table 1: Row 1). There is not a substantial difference between pure-vision and language-aligned vision models in predicting IT responses. In contrast, language models perform far worse than pure-vision and language-aligned vision models in prediction of early ventral stream activity (Table 1: Row 3). (The poor performance of these models in early visual cortex provides a convenient sanity check, demonstrating that the LLMs are not just statistically overpowered in general.)

**Vision versus Language in Macaque Ventral Stream** Applying the same encoding procedures with the same stimuli to prediction of brain activity in macaque visual cortex, we find that, as in humans, pure-language models are remarkably accurate in predicting high-level ventral visual activity (Figure 1B: Bottom Row, Table 1: Row 2). Also as in humans, we find that pure-language models perform poorly in prediction of early visual cortex (Table 1: Row 4). There is also no substantial difference between pure-vision models and language-aligned vision models. There is, however, a slightly more pronounced difference between the pure-language and pure-vision models in macaque IT (.343 versus .441) compared to human OTC (.332 versus .365).

**Rank-Order Correlation of Models across Species** The results above demonstrate that vision and language models perform largely similar across the species; but what about the more granular comparison instantiated by the *rank-order* correlation of models across the species (Figure 1B). For pure-vision models, this correlation is $\rho = 0.53$ ($p = 0.006$), suggesting individual vision models are similarly predictive of both species. For pure-language models, this correlation is 0.64 ($p = 0.009$), even higher than for vision models. In short, better language models of human brains are also better language models of macaque brains. Accordingly, hypotheses about why some language models do better than others in their prediction of human brains should apply in commensurate degree to macaque brains, as well.

Table 1: Human-Monkey-Model Comparisons Summarized

| Species | Region | Model Type | Encoding Score |
|---|---|---|---|
| | | | Group Mean [95% BCI] |
| Human | OTC | Vision | 0.365 [0.336, 0.395] |
| | | Language | 0.332 [0.298, 0.375] |
| | | Multimodal | 0.365 [0.336, 0.393] |
| Macaque | IT | Vision | 0.441 [0.36, 0.523] |
| | | Language | 0.338 [0.265, 0.42] |
| | | Multimodal | 0.415 [0.343, 0.488] |
| Human | EVC | Vision | 0.335 [0.302, 0.356] |
| | | Language | 0.333 [0.296, 0.355] |
| | | Multimodal | 0.333 [0.296, 0.355] |
| Macaque | V1 | Vision | 0.341 (1 Subject) |
| | | Language | 0.107 (1 Subject) |
| | | Multimodal | 0.348 (1 Subject) |

Table 2: Interspecies Unique Variance Analysis Summarized

| Regression | Unique | | | Shared (2-Way) | | | Shared (3-Way) |
|---|---|---|---|---|---|---|---|
| | Species | Vision | Language | Species+ Vision | Species+ Language | Vision+ Language | All Predictors |
| Human ∼ Macaque | ∼.0% | 18.4% | 14.6% | 8.3% | 12.1% | ∼.0% | 46.6% |
| Macaque ∼ Human | ∼.0% | 11.8% | ∼.0% | 11.5% | 25.4% | 13.1% | 38.2% |

**Unique Variance Analysis of Interspecies Difference** The slight difference in language versus vision models in predicting macaque IT relative to human OTC may be due to multiple factors, including the species-specific recording modalities and preprocessing steps (e.g., electrophysiology versus functional MRI). The question of most relevance, here, though, is whether the difference is attributable primarily to language (or at least, language as encoded in the pure-language models). To assess this directly, we performed a 3-way unique variance analysis, predicting human OTC activity

with all seven combinations of the groupings of three predictors: macaque brain activity, vision and language model embeddings (here the model embeddings for the most OTC-predictive model from each set).

The goal of this analysis is to understand how well different types of models predict 'uniquely human' neural signals in OTC. The logic being that if the difference between humans and monkeys is a difference attributable to language (over and above vision), then the unique variance explained by language models should be greater than the unique variance explained by vision models. We find this not to be the case (see Figure 2, and Table 2). The majority (8.3% + 12.1% + 46.6% = 67%) of the explainable variance in human OTC is variance that is shared with macaque IT and at least one of the models. We note there was almost no shared variance between the macaque and human data that was not also shared with one of the models. Of the remaining 'uniquely human' variance, a *roughly* symmetric amount is attributable to the language (14.6%) and vision (18.4%) models – though vision's unique variance is slightly higher. The relative symmetry of the unique variances suggests that the difference between human OTC and macaque IT is not a function uniquely of language, but may instead be a reflection of any number of the modes of difference in species, recording modality, or experimental setup.

**Summary** Here we show that the ability of 'language-aligned' vision models and pure-language models to predict image-evoked brain activity in human high-level visual cortex is likely not evidence of language having re-shaped vision. We find similar trends using these models to predict visual brain responses in a species that has no language, and demonstrate that those differences which do exist between humans and monkeys are not directly attributable to the structures of language, as captured by LLM embeddings, alone. Such is the nature of the 'monkey razor': If, *caeteris paribus*, an experimental effect holds in both humans and monkeys, that effect cannot be attributable to the structure, function, deployment, or learning of language *per se*. Thus, the more likely explanation is that the representational overlap between pure-language models and the high-level primate visual brain reflects a structure learnable in large part through the hierarchical encoding of natural image statistics. Language (as learned by language models) may approximate the representational endpoints of this process, but only to the extent that these statistics are reflected in the language we use to describe the world around us (a world the language models themselves cannot actually 'see').

The observations we have made in primate brain prediction align well with recent findings in AI research that show vision and language models seem to be converging on common – 'platonic' (17) – representations (18; 19), even in the absence of explicit alignment. And while we note that this particular alignment (between vision and language models) may be more a function of the effective overlap in the training data of the internet than of shared world knowledge, it does seem to suggest again that language reflects visual structure.

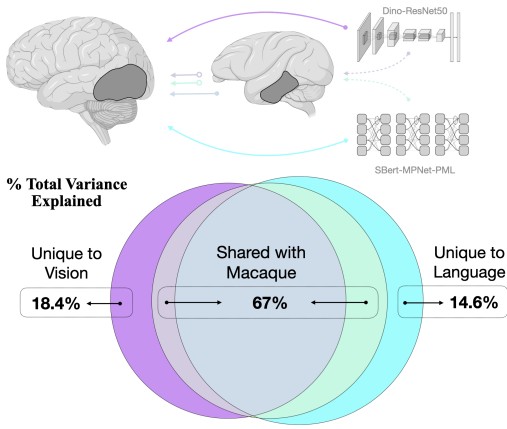

Figure 2: Results from a 3-way unique variance analysis designed to partial out the variance in human brains attributable uniquely to vision and uniquely to language (after partialing out the variance shared with macaque visual cortex). This figure shows the variance partitioning of human OTC across macaque IT, the most-predictive pure vision-model (Dino-ResNet50) and the most predictive pure-language model (SBERT-MPNet-PML). Most of the variance is shared across both macaques and models; the unique variance that is attributable to models is relatively proportionate across vision and language – with a slight advantage, perhaps, to vision as being the greater source of 'uniqueness' between the species.

Further work is needed to make sense of the lingering difference, however small, between language model predictivity of human OTC and macaque IT. One major factor that merits further scrutiny here is the translation between different neural recording modalities: fMRI signals, for example, may include later visual components (including feedback) not evident in the electrophysiological signals. Differences in experimental setup and task demands (i.e. stimulus duration, ISI, and freeviewing versus fixation) may also affect the extent to which semantic content is captured in visual cortex (20). In future work, we hope to assess the tradeoff between pure-language and pure-vision model encodings over time – an analysis that could unveil an even greater degree of similarity between humans and macaques than the initial similarity we've shown here. Perhaps more importantly, we could also aspire to collect or curate visual brain data in both species that pushes the limits of representation learnable through image statistics alone – and extends more explicitly into the kinds of conceptual territories where the structures of language are most indispensable for understanding.

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
