# OpenReview forum: "Monkey See, Model Knew: Large Language Models accurately Predict Human AND Macaque Visual Brain Activity"
_NeurIPS.cc/2024/Workshop/UniReps — UniReps_

### Official Review · Reviewer_opVb · 2024-09-30
**A strong foundational investigation into why we observe similarities in representations in language-based models and the human brain**

**Rating:** 10
**Confidence:** 5

**Review:**

This paper investigates a critical question about the role of language in representations in natural and artificial intelligence, capturing this workshop’s main goal by combining neuroscience and multimodal model research. The researchers detail the existing body of literature investigating how representations within language-based models can explain many language and visual representations within the human brain. These findings may suggest that the human visual cortex is linguistically-aligned, that language has shaped how visual information is processed and represented in the human brain. However, these authors point out that language at large may represent a statistical organization of information, when language representations align with visual representations, it may be due to a common hierarchical organization of the world in general. Building on previous investigations where features extracted from vision/language/multi-modal models are used to create encoding models to predict human brain activity, these authors extended predictions to brain activity in rhesus macaques, a primate species without formal language. If language-aligned models predict human brain activity because language restructured visual representation, the models should not be able to predict monkey brain activity. The authors found similar predictive power with both human and monkey encoding models. They strengthened their analysis by further investigating differences in encoding model performance, and statistically established the differences were unlikely uniquely due to language.

This work concisely builds off a growing amount of literature that compares neural representations in AI systems and human brains. It investigates, why we find large similarities in linguistic and visual representations, clarifying that language itself is likely not the foundation but a larger statistical organization of how the brain perceives the world. Without foundational work like this, our future understanding of representations in both the human brain and multimodal models would be restricted by our assumptions. The authors emphasize the need for more investigation into the language model predictivity between human and monkey brain regions. Deeper findings in this realm could contribute to our understanding of intelligence as a whole. For example, answering if differences in information representation can explain behavioral differences between species. I believe this work could be extended following past literature in the field. Investigating specific semantic categories of representations (people, places, things, and even more fine-tuned concepts) to see if they appear different in language-having primates vs non-language primates could have interesting results from a broader nature vs nurture perspective.

Overall, the writing is very concise and understandable. The authors do a great job of explaining past findings without going into too much detail. The authors acknowledge limitations in their experiment (i.e., comparing different neural recording methods) and provide great detail of their results with statistical thresholds and calculations.

I believe this paper would make a great contribution to the UniReps workshop. The authors have put forward great writing, helpful visualizations, and clear communication of important science.

---

### Official Review · Reviewer_wWZk · 2024-10-03
**Very interesting premise but lacks in succinct communication of aims, experimental setup and results.**

**Rating:** 3
**Confidence:** 4

**Review:**

Summary: The paper seeks to address the debate about the role of language in shaping visual system of humans, which has been strengthened by work on LMs being able to predict brain activity by comparing ability of LMs to predict non-human brain activity. Based on findings, it concludes that since LMs also predict macaque brain activity (who don’t have language), LM predictive powers are not due to vision being-language aligned but because it reflects underlying statistical structure of the word.

Strengths:

1. The work is based on an interesting premise of whether species with no language (such as macaques) can also have representations that align with the LMs of today.

2. The experiment results presented are statistically sound.


Weaknesses:

1. The argument of the paper is set in very convoluted terms. The main goal is unclear- is the work a verdict on whether language has shaped vision? Or does it aim to show that representational alignment is indicative of nothing? Or does it aim to show that vision and language all reflect a common underlying structure of the world?  The hypothesis must be made much more clear.

2. The abstract mentions “underlying statistical structure” which in itself is an interesting thought but the work does nothing further to address it and there are no elucidations on what this means or could mean for representational analysis?

3. There is no discussion of models available- today LMs, LLMs, VLMs, can mean a wide range of things so what model architectures and types are we talking about? The only clue about models is from the extremely small font in the figure and there is no discussion or no appendix about what are the multimodal models here. The conclusions might very well be limited to some architecture or training types but there is no way to determine that.

4. The experimental setup for LMs is also relatively unclear- what were the prompts for processing input? What were the instructions given?

5. The paper talks of common beliefs or hypothesis in the field but there are very limited references on these- not in the main paper and there is no appendix either. It would be much more clear to have a related work section and how this work synthesizes its claims from those and tries to address them.


Corrections: Referencing format is not Neurips 2024.

---

### Official Review · Reviewer_px4j · 2024-10-04
**Great study; comparison of similar recording modalities would strengthen the findings**

**Rating:** 6
**Confidence:** 3

**Review:**

Originality, Significance & Theoretical Merits:

The authors tested whether language-based machine learning (ML) models can predict image-evoked brain activity (using image captions) in monkeys, in a similar way to how these models predict image-evoked neural activity in the human brain. This is a valuable test, as monkeys do not use language, and their brains have not evolved to represent language-based concepts. The study could challenge the interpretation of previous findings that language models can predict neural activity in the human visual cortex, which some have attributed to language shaping the visual cortex during human evolution. The experiments are novel, and the findings will likely be of broad interest to the neuroscience community, particularly for researchers using deep neural networks (DNNs) to predict neural activity across the brain.

However, it is unclear whether new data was collected for this study. The authors reference the following study regarding the images used:
Emily J. Allen, Ghislain St-Yves, Yihan Wu, et al. (2022). A massive 7T fMRI dataset to bridge cognitive neuroscience and artificial intelligence. Nature Neuroscience, 25(1):116–126. doi: 10.1038/s41593-021-00962-x.

While it appears that the fMRI data comes from this study, the source of the monkey electrophysiology data is unclear (apologies if I missed this in the manuscript). Could the authors explicitly state the origins of all datasets used?

Technical Soundness:

While the chosen dataset, models, and experimental design are sound, comparing fMRI data to multi-unit activity might not be ideal. This comparison assumes that the difference between the two recording modalities can be captured by a linear encoding model (I assume the authors used a simple linear encoding model). Still, the explained variance for the language models predicting monkey data is interesting in itself, as it suggests significant predictability without implying that language shaped these brain areas.

Clarity:

The manuscript is generally clear, but several details are missing:

	•	What does NCSNR stand for, and what is its significance?
	•	On line 76, there is a reference to “Figure 1?”.
	•	I am confused by what Table 1 is showing. In the main text, it is referenced only in relation to the human data analysis, but it also seems to contain results for monkey data. Furthermore, the values described for the monkey data in the results section differ from those in Table 1. Additionally, in the human early visual cortex (EVC) section, both Language and Multimodal models have the same value of 0.333, which is surprising, given that the authors describe this as a low value for language models in the main text.
	•	There is a typo in the reference to Table 2: the word “Table” is missing.

Additional Comments:

	•	Figure 2 does not add much information beyond what is already described in the text. The figure caption includes some details that are missing from the main text, but the figure itself only presents three variance values, which are also mentioned in the text.

---

### Decision · Program_Chairs · 2024-10-10

**Decision:**

Accept

**Comment:**

In light of the positive reviewers' feedback and relevancy of the submission, we are pleased to accept this paper for presentation at UniReps 2024. We kindly ask the authors to incorporate the reviewers' suggestions and feedback in the final camera-ready version of the manuscript.